# Effect of Grain Refinement and Dispersion of Particles and Reinforcements on Mechanical Properties of Metals and Metal Matrix Composites through High-Ratio Differential Speed Rolling

**DOI:** 10.3390/ma13184159

**Published:** 2020-09-18

**Authors:** Ahmad Bahmani, Woo-Jin Kim

**Affiliations:** 1School of Metallurgy and Materials Engineering, College of Engineering, University of Tehran, P.O. Box 14395-515 Tehran, Iran; ahmadbahmani@ut.ac.ir; 2Department of Materials Science & Engineering, Hongik University, 72-1 Mapo-gu, Sangsu-dong, Seoul 121-791, Korea

**Keywords:** ultrafine grains, severe plastic deformation, differential speed rolling, mechanical properties, high strength, texture

## Abstract

A differential speed rolling (DSR) technique that provides capability of producing large-scale materials with fine grains and controlled texture in a continuous manner has attracted several researchers and industries. In this study, we tried to review the articles related to DSR and especially the high-ratio DSR (HRDSR) technique that is associated with a high speed ratio between the upper and lower rolls (≥2) and compare the change in microstructure and mechanical properties after HRDSR with the results obtained by using other severe plastic deformation (SPD) techniques to see the potential of the HRDSR technique in enhancing the mechanical properties of metals and metal matrix composites. The reviewed results show that HRDSR is an important technique that can effectively refine the grains to micro or nano sizes and uniformly disperse the particles or reinforcement throughout the matrix, which helps extensively in improving ambient and superplastic mechanical properties of various metals and alloys.

## 1. Introduction

Microstructural parameters such as grain size, texture, dislocation density and second phases determine the materials properties and they are changed by the processing techniques and conditions [1,2,3]. Therefore, understanding the effect of microstructural parameters reveals the progressing route for new materials with advanced properties. Moreover, it is worth knowing all the capabilities and limitations of various processes to achieve the desired microstructure for specified mechanical properties. Processing types, such as casting, forming and heat treatment, dictate the desired microstructure and hence material properties. 

Severe plastic deformation (SPD) is an applicable process that can be used to effectively change the mechanical properties of materials. This process is a general term of metalworking, denoting any process that applies large strain to materials at low and intermediate temperatures, resulting in an ultrafine grain (UFG) (≤1 μm) or even nanocrystalline (NC) structure (≤100 nm) [4].

Based upon applications and capabilities, several SPD techniques have been invented, such as equal channel angular pressing (ECAP) [5,6,7,8], high pressure torsion [9,10,11], three roll planetary milling or screw rolling (SR) [12,13,14,15], accumulative roll bonding [16,17], multi directional forging [18,19,20,21] and asymmetric rolling (ASR) [22] or differential speed rolls (DSR) [23,24,25,26,27,28,29,30]. However, for the production of ultra-fine equiaxed grains, severe, complex and high stress is required, and therefore material size is limited for most of the techniques. Nevertheless, DSR has a considerable advantage in producing long sheets with ultra-fine grains, possibly in a continuous manner. In fact, DSR is a modification of the generic rolling process where the upper and lower rolls rotate with different speeds [24,31].

Differential speed can be applied to the rolling process through various ways, such as different materials (that induce different friction speed to rolls), diameter or rotational speeds (Figure 1a–d). The most important characteristic of DSR process is the speed ratio coefficient, SR, which is defined as the speed ratio of the upper to lower roll. In fact, the level of rolling asymmetry can be represented by SR. DSR was first theoretically described in 1941 [32], and after that several technical improvements were applied to this rolling process [24].

Kim et al. [33,34,35,36] introduced a developed DSR technique, which is called high-ratio differential speed (HRDSR), by applying a large shear strain during asymmetric rolling with a high contact friction where the roll-speed ratios (SR) are equal and larger than 2 and it resulted in quite effective grain refinement. This method not only has the ability to refine the grain size significantly in a few passes, but also it is applicable to hard-to-deform materials, which is a limitation of the majority of SPD processes [37]. HRDSR can be run in a continuous manner, which makes it suitable for large-scale manufacturing. The layout of the prototype machine for the continuous HRDSR process is presented in Figure 2, where the metal sheets on the coiler are uncoiled, HRDSR-processed (HRDSRed) and coiled in the second coiler/uncoiler [19]. In the continuous HRDSR, this process is repeated several times to achieve the final thickness. Comparing the outcoming sheet from HRDSR and the equal speed rolling (ESR), the sheet after HRDSR has a higher quality in flatness and shape when a large thickness reduction per pass is applied. The HRDSR-processed sheet is flat and smooth while the ESR-processed sheet is wavy and uneven (Figure 2d). Pesin et al. [38,39] showed in the finite element method (FEM) simulation that high shear strain through thickness of the sheet can only be achieved in HRDSR when the friction coefficient between the sample and the surface of the rolls is high, but the high friction can cause material damage and roll wear. Through control of the amount of thickness reduction per pass and the addition of a small amount of lubricant; however, there have been efforts to minimize these problems in the continuous HRDSR mill (Figure 2c).

Many articles have been published on HRDSR and unique results have been obtained through this process and thus it is important to review the related articles to understand the mechanisms improving the mechanical properties. Here, the microstructures and mechanical properties of various metals and metal matrix composites, which have been processed by HRDSR, have been reviewed and analyzed.

## 2. Method of Imposing a Large Plastic Deformation in HRDSR

### 2.1. Effect of Roll-Speed Ratio on Microstructure

The speed ratio (SR) is a key parameter in determining the microstructure and therefore the properties of the alloys. In an attempt to understand the effects of various speed ratios on the microstructure and mechanical properties of metals, Kim et al. applied different speed ratios (1 to 3) applied on the Mg-3Al-1Zn (AZ31) magnesium alloy with the same reduction ratio of 20% through a single pass at 513 K [40]. Figure 3a–f depicts the image quality (IQ) maps from electron back-scattered diffraction (EBSD) observed at the middle of the longitudinal cross-sections of the rolled alloy. The grain size and fraction of high-angle grain boundaries (HAGBs, *θ* > 15°), which are analyzed with the aid of orientation imaging microscopy software (TSL-OIM), as a function of speed ratio has been drawn in Figure 4. The value has been obtained based on the average of the three grain sizes measured at the top, middle and bottom layers of each sample by means of EBSD analysis. The fraction of HAGBs reduced from 0.89 to 0.61 at SR of 1.0 and 0.54 at SR of 1.1. However, the fraction of HAGBs (0.81) for SR of 3 was restored to that of initial state (0.89). Moreover, after rolling with SR of 3, the average grain size decreased from 9.0 to 5.9 μm. Continuous dynamic recrystallization (CDRX), which usually takes place during the SPD process, proceeds slowly at low values of SR (1.0 or 1.1) and thus the microstructure contains a high fraction of low-angle grain boundaries and cell boundaries. At high SR values (2.0 or 3.0), however, high stored lattice strain energy, high-density dislocations and the high adiabatic heat generated during heavy plastic deformation accelerate the CDRX process, resulting in the formation of a large amount of high-angle grain boundaries and grain refinement.

For achieving ultrafine or nano-sized grain size from metals and metallic alloys, a total thickness reduction higher than 60% is often required at SR ≥ 2 at low temperatures [41]. Figure 5a–d show the TEM micrographs and the corresponding selected area diffraction (SAED) patterns of the pure Ti deformed by HRDSR with a single pass for a reduction of 63% at SR of 1, 3, 4 and 5 at room temperature [41]. At an SR of 1, deformation twins and diffuse deformation bands were observed. At an SR of 3, the elongated cells formed along the rolling direction. At an SR of 4, many cell walls were converted to sharper subgrain boundaries and the elongated structure was divided up into smaller domains. At an SR of 5, the microstructure was transformed into more equiaxed grains with higher misorientations. Within some of the grains (marked by an arrow), nanosized subgrains (a size of ∼0.1 μm) could be located. These microstructural characteristics at different SRs indicate that CDRX is accelerated as SR is increased. This results because a higher SR produces a more severe plastic deformation under the same thickness reduction.

Polkowski et al. [42] also observed an increase in the fraction of HAGBs in pure Cu with an increase in SR (from 1 to 4). In addition, they reported a significant increase in the fraction of the shear texture component and occurrence of extensive cross slip due to the imposition of high shear stresses during HRDSR. Yu et al. [43] carried out the asymmetric cryo-rolling process on the Al 1050 alloy and reported achievement of nanosized grains. Figure 6 shows the TEM micrographs of the samples after the rolling process. Compared to the grains obtained at SR of 1.1, the grains obtained at SR of 1.4 are smaller in size (211 nm vs. 360 nm) and more equiaxed in morphology. This result demonstrates that the combined effect of low processing temperature and high SR produces a synergy effect on grain refinement. Very recently, Han and Kim [44] showed that application of HRDSR to a Ni-rich NiTi alloy at a subzero temperature (268 K) produced a nearly fully amorphous microstructure and the nanograin refinement through subsequent annealing improved superelasticity of the alloy.

### 2.2. Principle of HRDSR

Figure 7 shows the grid distortions and neutral points for ESR and HRDSR processes. To understand the reason why HRDSR produces a significantly higher grain refinement effect compared to ESR and why a lower roll force is imposed during rolling, such that a large thickness reduction per pass is possible and a hard-to-deform material can be easily deformed, Ji et al. [45] performed the FEM studies. As shown in Figure 7a,b (for a case of thickness reduction of 70% with a single pass under ESR and HRDSR) in ESR, neutral points are located at the top and bottom surfaces of the sheet and they are positioned at 9.83 mm from the entrance of the sheet for deformation. In HRDSR, the neutral points at the top and bottom surfaces of the sheet are located at different positions. They are at the distances of 13.06 and 3.16 mm away from the entrance of the sheet. During the HRDSR process, the vertical lines of the grid are inclined significantly in the same direction (to the rolling direction) throughout the thickness, indicating that HRDSR produces a higher and more uniformly distributed shear strain throughout the thickness of sheet sample compared to ESR [45].

Figure 8 compares the strain along the thickness of sheets produced by ESR and HRDSR techniques for a thickness reduction of 70% with a single pass at SR = 3 [46]. The shear strain here has been calculated based on the strain increments. The shear strain is zero in the middle of ESR where it is about 3.5 in the case of HRDSR sheet. The shear strain is high in all points of HRDSR and moreover the difference from the middle to the edge is small. As HRDSR is more effective in applying shear strain than ESR, the effective strain accumulated during the rolling process is larger in HRDSR than in ESR. This is shown in Figure 9, where the effective strain accumulated during rolling is monitored as a function of processing time [47]. Furthermore, the accumulated strain is more uniformly distributed along the thickness of the sample during HRDSR. For example, the effective strain ranges from 3.15 to 3.34 in HRDSR (with SR = 3), while it ranges from 1.43 to 2.36 in ESR. The higher shear strain applied on the samples makes the sheet store more strain energy and thus produce more effective grain refinement, and the smaller difference in strain along the thickness direction results in the formation of more homogenously refined microstructure.

Figure 10 shows the effect of SR on the roll stress, which is monitored as a function of processing time, under a thickness reduction of 70% by a single pass [47]. The maximum roll stress tends to decrease with the increase in SR. At SR = 3 (HRDSR), it is approximately six times smaller than that calculated at SR = 1 (ESR). Figure 11a schematically shows the deformation of a square element after deformation through ESR, DSR, and HRDSR. The deformed shape of the element after processing illustrates that the principal stress direction of the material in HRDSR is rotated from the principal stress direction in ESR and the final shape is more deformed geometrically. When the same yield condition is assumed, the principal stresses, σ_1_ and σ_2_, and the principal direction are equal for ESR, DSR and HRDSR. However, the pressure direction applied on the surface of the sheet is different from the principal direction. The amount of pressure on the surface perpendicular to the rolling direction can be estimated using Mohr’s circle for stress (Figure 11b). As the SR value increases, the shear stress increases and thus the angular difference between principal and current stresses increases (such as 2θ_1_ and 2θ_2_). The roll pressure, which can be determined from σ_y_ value, tends to decrease with the increasing the angular difference. Furthermore, a lower hydrostatic stress develops in HRDSR than in ESR [48], making the pressure applied on the surface of the sheet further decrease in HRDSR.

Above analysis and FEM simulation results explain why the roll pressure is considerably low in HRDSR, such that a large thickness reduction per pass is possible in HRDSR and the hard-to-deform materials are not easily breakable during severe plastic deformation. This unique advantage of HRDSR has been used in fabricating a thin open-cell Ni foam sheet with a high specific strength, high hardness and moderate porosity [49]. As shown in Figure 12, more curved cell shapes with smaller curvatures can be observed with increase in SR due to more shear stress on the cell ligaments. As the SR increased, the accumulated dislocation density was increased and hence grain refinement in the cell walls was promoted through accelerated CDRX. Due to a great reduction in roll pressure during HRDSR, the porosity decreased from 90% to 25% after ESR, while it decreased less (to 36%) after HRDSR with a SR of 3 [49].

Even though shear strain is the key issue in refining the grain structure in HRDSR, it can cause more serious material damage compared to ESR due to additional shear strain (Figure 13). Persin et al. [38] showed that the damage parameter *D* (defined based on the Cockroft–Latham model) is very high at the bottom and edge of the sheet in HRDSR due to the development of high tensile stress at the corresponding locations. Therefore, optimization of contact friction, thickness reduction per pass, the roll-speed ratio, the roll-speed and temperature, the initial thickness of the sheet material, and the sheet temperature is important for producing an ultrafine-grained sheet with smooth surface and uniform thickness.

## 3. Enhancement of Mechanical Properties and Microstructures by HRDSR

### 3.1. High Strength and High Ductility

Grain boundaries as a source of crystallographic defects have an important effect on the majority of the properties of the materials. They are sites of hindering dislocations, and dislocations need to change their slip system after arriving at the grain boundaries. Therefore, more energy or stress is required to move the dislocations across the grain boundaries and hence the strength of materials increase with the increasing density of grain boundaries.

It has been well established that grain size is inversely related to the yield strength through the Hall–Petch equation [20,50]:(1)$$\sigma_{y}=\sigma_{0}+Kd^{-0.5}$$
where $\sigma_{y}$ is the yield stress, $\sigma_{0}$ is the friction stress, *d* is the average grain size and *K* is the stress-concentration factor. The main mechanism that has been mentioned to explain the Hall–Petch theory is based upon the pile-ups of dislocations, which are blocked on grain boundaries [50]. Both $\sigma_{0}$ and *K* values are important in determining the materials strength. *σ*_0_ is related to the composition and crystal structure. However, Armstrong et al. [51] showed that *σ*_0_ is also related to the crystals orientation as follows:(2)$$\sigma_{0}\;=\;M\tau_{0}$$
where *M* is the Taylor orientation factor and τ_0_ is the critical resolved shear stress (CRSS) for an operative slip system. The *K* constant in the Hall–Petch equation is related to the material composition, texture and deformation mechanism. For a given texture, the relationship between yield strength in a certain orientation and *d*^−1/2^ follows the Hall–Petch equation with a good agreement. However, the *K* value changes if the texture or loading direction is changed. In this case, loading direction plays an important role in the *K* value and it has been well established that loading along processing direction (usually perpendicular to the dense planes) results not only in higher *K* values but also higher yield strength [52]. Therefore, it can be argued that both factors of *K* and *σ*_0_ are related to the crystallographic texture of materials [50,53]. This effect of texture on yield strength is especially important in Mg alloys with hexagonal close-packed crystal structure, having a limited number of slip systems.

Kwan and Kim [54] proposed a Hall–Petch equation for Mg alloys, which simultaneously considers the effects of both grain size and texture on strength of Mg alloys using the Schmidt factors (SFs) for basal slip (*SF*):(3)$$\sigma_{y}=\sigma_{0}+Kd^{-0.5}=\frac{26.6}{SF}+\frac{41.7}{SF}d^{-0.5}$$

Equation (3) predicts that *σ*_0_, *K* and *σ_y_* decrease as the SF for basal slip increases. Since the SF values for the equal-channel-angular-pressed (ECAPed) Mg alloys are typically 0.3~0.35 [55], the *σ*_0_ and *k**_y_* values for the ECAPed Mg alloys will be 76~88.7 MPa and 119.1~139.0 MPa μm^1/2^, respectively. The SF value for the extruded Mg alloys are 0.20~0.22 [55,56] and thus, the *σ*_0_ and *k**_y_* values will be 120.9–133.0 MPa and 189.5–208.5 MPa μm^1/2^, respectively. The SF values for the rolled Mg alloys, which are smallest, are 0.16–0.18 [57,58] and hence the *σ*_0_ and *k**_y_* values of rolled Mg alloys will be 147.8~166.3 MPa and 231.7~260.6 MPa μm^1/2^, respectively.

Kim et al. [35] applied HRDSR to AZ31 magnesium alloy sheets and found that when the AZ31 alloy were rolled with a large thickness reduction of about 70% at low temperature (423 K), ultrafine grains with strong basal texture could be obtained. Due to this, as shown in Figure 14, there is an obvious discrepancy in *σ*_0_ and *K* values between the HRDSRed samples (including the extruded and ESRed samples) and the ECAPed samples. Significantly reduced yield strengths were obtained for the same grain size in the samples processed by ECAP, showing lower *σ*_0_ and *K* values due to increased SF by rotation of basal planes toward the orientations for favorable slip. Super high strength has been obtained from the AZ31 Mg alloy using HRDSR through this combined effect of grain refinement and texture control, showing the potential of this process for achieving very high strengths from Mg alloys [35].

With the aid of DSR or HRDSR combined with various heat treatments, a wide range of properties can be achieved. Huang et al. [59,60] showed that DSR at very high temperatures (i.e., 793–832 K) can effectively reduce the basal texture of AZ61 alloy by suppressing tensile twinning activity during rolling and hence improves the tensile ductility. This method is, however, deleterious for yield strength. Therefore, this method is applicable when ductility is more important than strength. To improve the ductility of ZK60 alloy, ZK60 alloy was hot rolled (at 823 K) or hot/warm rolled (at 823 K and 503 K) at the SR of 1.2 and then annealed [61]. Figure 15a shows the tensile test results of the ZK60 alloy subjected to various deformation-heat treatment processes representing various properties with high ductility and strength. The hot rolled plus annealed alloy and hot/warm rolled plus annealed alloy exhibit higher ductility (24.8% and 25.4%) compared with the warm rolled plus annealed alloy (20%) due to enhanced work hardening caused by basal texture weakening (Figure 15b). As a result of texture weakening, this alloy was found to be highly deformable in tension at room temperature. Aging after the hot/warm rolling plus annealing could enhance the strength that considerably decreased due to the effect of basal texture weakening. These results show the possibility of producing Mg alloy sheets with high formability at room temperature thanks to the texture weakening, and high strength gained through using the aging process after forming of the texture-weakened Mg sheets. Huang et al. [62] showed that the DSRed AZ31 Mg alloy sheet with weakened basal texture exhibited a larger tensile elongation, a smaller Lankford value, a larger strain hardening exponent and improved Erichsen values at room temperature. The HRDSR-processed AZ31 Mg alloy sheet, on the other hand, exhibited a significant improvement in the limit dome height by 46% compared to the commercial AZ31 sheet at 523 K, mainly due to grain refinement effect [37].

The effect of HRDSR on grain size and mechanical properties of various materials has been given in Table 1, showing the capability of this technique in refining the grains and improving the mechanical properties of various materials. As it can be seen, material strength including yield and ultimate tensile strength was extensively increased after HRDSR.

A comparison between the influence of HRDSR and other processing routes on the mechanical properties of pure Ti has been given in Table 2. A very high strength Ti was obtained by HRDSR, which is 804 and 915 MPa for yield and ultimate tensile strength, respectively, with 19.0% elongation. Comparing to the majority of the mechanical properties that have been obtained using other processing routes, it is one of the highest strengths with good ductility. However, with double route deformation processing (ECAP+cold rolling), which was carried out by Stolyarov et al. [75], higher strength is obtained, which is due to more effective grain refinement through extremely severe plastic deformation. This process is, however, more complex and less practical for mass production of ultrafine-grained sheets compared to the HRDSR process.

### 3.2. Enhanced Superplasticity

Grain refinement through HRDSR not only enhances the ambient mechanical properties, but also improves superplasticity at elevated temperatures. Superplasticity is the ability of a polycrystalline material to experience a very high elongation in tension without necking. The tensile elongation for superplasticity is generally considered above ∼ 400% [80]. The strain rate sensitivity (*m*) of a variety of Mg alloys with small grains has been obtained with ~0.5 [81,82], implying that grain boundary sliding (GBS) is a rate-controlling deformation mechanism and is responsible for superplasticity. For superplasticity, in addition to fine grain size, the presence of densely and uniformly distributed secondary phase particles with high thermal stability is important in suppressing grain growth during superplastic deformation. 

Figure 16 shows that the microstructures of three different zones containing top, middle and bottom of the HRDSRed ZK60 alloy have almost the same grain size with same strong basal texture [83]. The observed traces of a dense micro-scale shear bands (black thick lines indicated by white arrows) indicates that grain refinement occurred via CDRX within shear bands. As shown in the inset in (c), very fine precipitate particles (MgZn_2_) densely cover the boundaries of ultrafine grains and pin them.

Figure 17a–d show the microstructure of as-cast Mg–9.25Zn–1.66Y alloy and the deformed microstructures after ESR and HRDSR with SRs of 2 and 3. In the cast microstructure, the eutectic phase with lamellar structure composed of icosahedral (*I*) phase and α-Mg phases is almost continuous along the grain boundaries. During ESR, the eutectic structure was refined to some extent, but its continuous network structure remained integrated. However, the eutectic structure was fragmented intensively during the HRDSR process. The eutectic network structure was destroyed, and the broken pieces were aligned to the rolling direction. The eutectic *I*-phase structure was more effectively broken into a lot of smaller particles and the broken pieces were more uniformly dispersed into the matrix when a higher SR was applied. The above two results demonstrate the capability of HRDSR in crushing and dispersing the secondary phases, as well as grain refining.

Superplasticity of the HRDSRed Mg alloys with finely and uniformly dispersed Mg_17_Al_12_, MgZn_2_, (Al,Mg)_2_Ca and *I*-phase particles [85,86,87,88,89] is compared with that of the Mg alloys subjected to SPD by ECAP [90,91]. This is shown in Figure 18. The comparison was made at a given strain rate of 10^−3^ s^−1^. The following results are observed from the plot. Firstly, the HRDSR processed Mg alloys show superplasticity almost equivalent to that exhibited by Mg alloys processed by ECAP. Secondly, depending on the chemical composition (i.e., the type and amount of secondary phase), the degree of superplasticity achieved from the Mg alloys processed by HRDSR is different. Some HRDSRed Mg alloys exhibit good superplasticity in the temperature range of 473–523 K, while some HRDSRed Mg alloys do in the temperature range of 523–573 K. It is usually expected that increasing temperature and decreasing strain rate increases superplasticity, but poor superplasticity is observed above 573 K. This is because ultra-fine grains became unstable above 523 K [87]. Simultaneous achievement of low-temperature superplasticity below 0.5*T_m_* (*T_m_*: melting temperature) and high-strain-rate superplasticity (HSRS: superplasticity at strain rates above 10^−2^·s^−1^) above 0.5*T_m_* from the identically processed material is advantageous for industrial application of superplastic forming technique, but achievement of HSRS above 0.5*T_m_* is difficult due to the instability of ultrafine grains at high temperatures. Recently, Lee and Kim [92] showed that by controlling the speed ratio in HRDSR, one can develop a microstructure that is stable at high temperatures above 0.5*T_m_* in Mg–Y–Zn–Zr alloys by controlling the particle size and distribution through using different SR. The properly processed alloy exhibited excellent HSRS at 673 K (~700 % at 10^−2^ s^−1^).

### 3.3. Metal Matrix Composites with Ultrafine Grains and Uniformly Dispersed Reinforcements 

The effective shear strain, which is applied through HRDSR, not only refines the grain size of metals and alloys but also can effectively disperse the micro- and nanosized reinforcements in the metal matrix composites, which is critically important for the strengthening of the metal matrix composites. An attempt for the development of an in-situ aluminum matrix for compositing with TiC as reinforcement has been carried out by Kim et al. [46], where the cast composite was subjected to several steps of conventional rolling (ESR) and then HRDSR. As can be seen in Figure 19, conventional rolling was not so effective in the dispersion of the clusters of TiC particles, even after 14 passes (Figure 19c), but HRDSR resulted in effective particle distribution, even after a single (Figure 19d) and especially after two passes (Figure 19e). These well-distributed particles have resulted in effective improvement in strength and hardness. 

In an attempt to prepare multilayer graphene (MLG)/copper composites and carbon nanotube (CNT)/copper composites using a combination of ball milling and rolling (ESR or HRDSR), different amounts of CNT or MLG as a reinforcement were added to copper powder and rolled by using a sheath rolling technique [74,93]. The fabrication procedures and the microstructures of the composites are shown in Figure 20. It was found that the yield strength and strain hardening exponent (*N*) were significantly increased after adding more MLG through HRDSR but there was no significant change after adding more MLG through ESR (Figure 21). This effect shows that HRDSR is much more effective in breaking up and dispersing MLGs into matrix compared to ESR. The strengthening mechanism of MLGs through HRDSR could be explained in terms of the Orowan strengthening mechanism by the nanosized MLGs dispersed within grain interiors while CNTs’ main role in the Cu composites prepared by HRDSR was grain refinement due to the break up and uniform distribution of CNTs along to the rolling direction [74,93]. It was also reported [94] that sheath rolling of an aluminum alloy using CNT + TiC by HRDSR can strongly enhance the mechanical strength, because CNT and TiC particles can be well dispersed in the ultrafine-grained matrix. Figure 22 shows the HRDSRed aluminum matrix composite where the well-distributed TiC particles are surrounded by CNTs. Recently, it was demonstrated that carbon black (0–0.08 wt.%) and AZ31 Mg chips could be used to fabricate carbon black-reinforced magnesium matrix composites with a combination of extrusion and HRDSR [95]. The strength analysis indicated that uniform dispersion of 0.03 wt.% carbon black into the matrix could be achieved with the aid of strong shear flow generated by HRDSR.

## 4. Summary and Conclusions

HRDSR is a SPD rolling process that induces a large shear strain during thickness reduction. The key control parameter in HRDSR is the roll-speed ratio between the upper and lower rolls (≥2). Depending on the roll-speed ratio, friction condition, processing temperature and thickness reduction per pass, the texture and microstructure of materials can be controlled.As the accumulated effective strain in HRDSR is significantly larger than that in ESR and a much smaller lower roll force is imposed on the material in HRDSR than in ESR, ultrafine grained microstructure can be obtained in HRDSR with less effort. More homogeneously refined microstructures are also obtained in HRDSR compared to ESR, due to a more uniformly distributed strain along the thickness direction during rolling.Experimental results show that HRDSR is effective in improvement of strength/ductility and superplasticity through grain refinement, texture control and fragmentation and dispersion of secondary phase. HRDSR is also effective in fabricating metal matrix composites with high performance because the dispersion of reinforcement and grain refinement can be simultaneously and greatly enhanced by high shear flow induced during HRDSR.HRDSR has a high commercial application potential due to the possibility of scaling the product up to commercial dimensions and continuous production of the product, but there are many parameters (such as contact friction, thickness reduction per pass, the roll-speed ratio, the roll-speed and temperature, and the sheet temperature) that need to be optimized to achieve the texture controlled ultrafine-grained sheets with small surface roughness and uniform thickness distribution.

## Figures and Tables

**Figure 1 materials-13-04159-f001:**
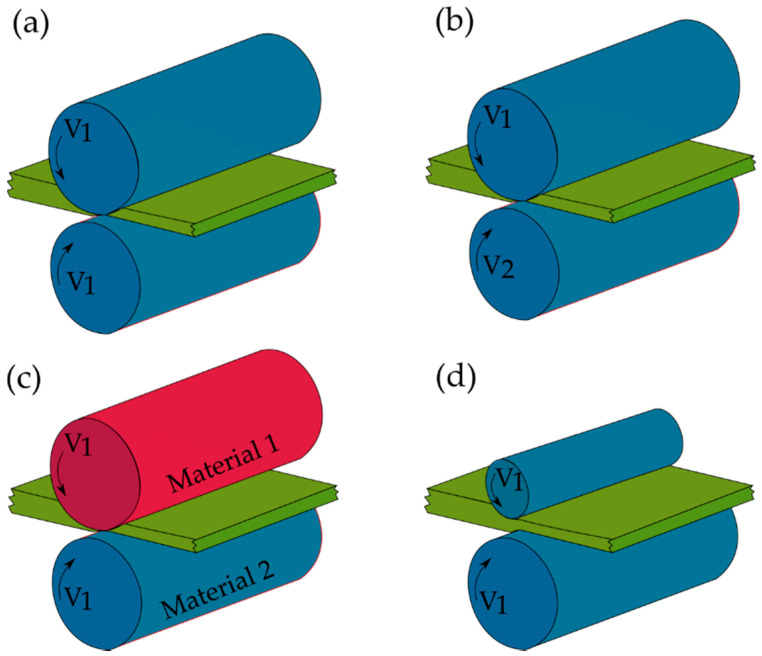
A schematic representation of (**a**) normal symmetric rolling (equal speed rolling) and various asymmetric rolling including (**b**) differential speed rolling, (**c**) different materials and (**d**) different roll sizes.

**Figure 2 materials-13-04159-f002:**
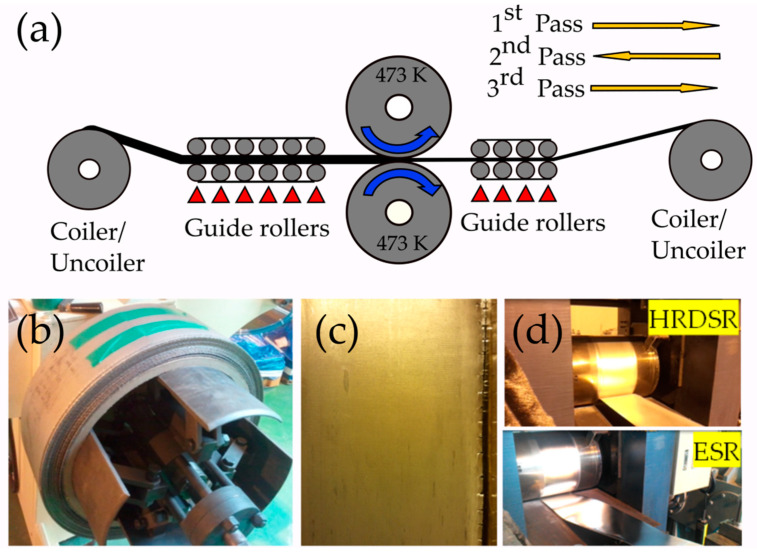
(**a**) The schematic of continuous high-ratio differential speed rolling (HRDSR) mill. (**b**) The coiled AZ31 magnesium alloy sheet processed after HRDSR and (**c**) the appearance of HRDSR-processed sheet and (**d**) HRDSR and equal speed rolling- (ESR)-processed AZ31 magnesium alloy sheets coming out of the rolls [37]. Reproduced with permission from Kim et al., *Mater. Sci. Eng. A*; published by Elsevier, 2014.

**Figure 3 materials-13-04159-f003:**
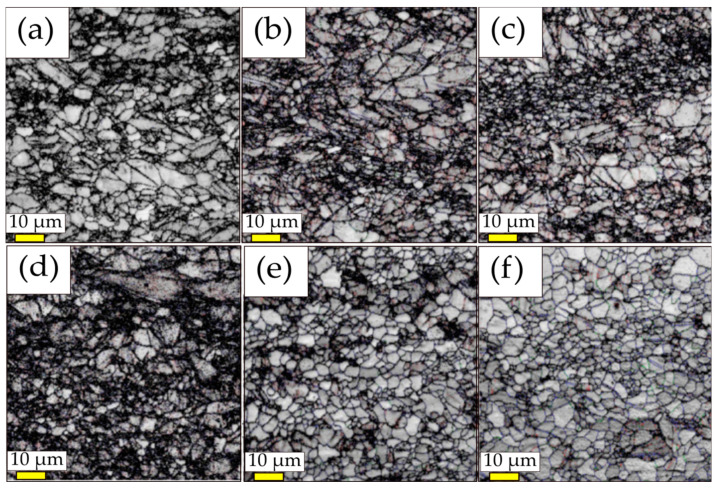
The electron back-scattered diffraction (EBSD) image quality (IQ) maps at the middle of longitudinal cross-sections of the sheet (rolling direction (RD) is the normal direction of figures plane) with speed ratio (SR) of (**a**) 1.0, (**b**) 1.1, (**c**) 1.2, (**d**) 1.5, (**e**) 2.0 and (**f**) 3.0 [40]. The speed ratio between the upper and lower rolls varied from one to three with the speed of lower roll fixed as three rpm (roll diameter of 400 mm). Reproduced with permission from Kim et al., *J. Alloys Comp*; published by Elsevier, 2011.

**Figure 4 materials-13-04159-f004:**
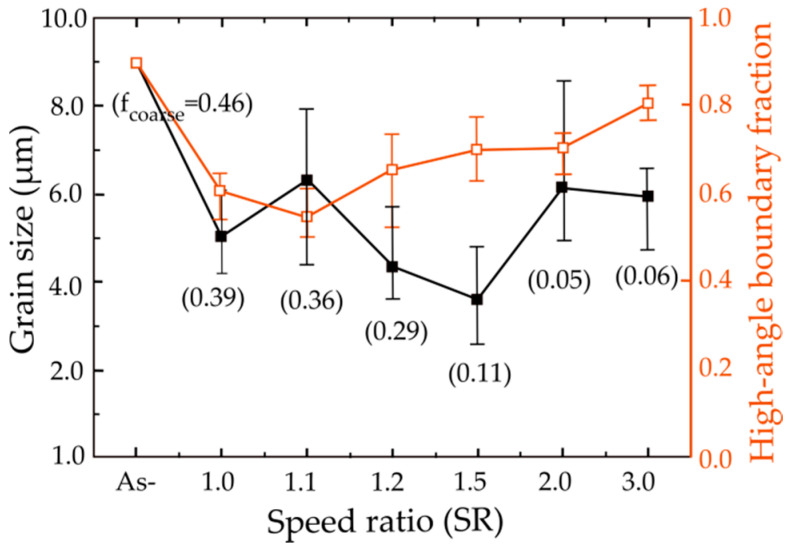
The average grain sizes and the average fractions of HAGBs, which were measured at the top, middle and bottom layers of the samples deformed at different SRs [40]. f_coarse_ is related to the fraction of grains that have not fully refined. Reproduced with permission from Kim et al., *J. Alloys Compd.*; published by Elsevier, 2011.

**Figure 5 materials-13-04159-f005:**
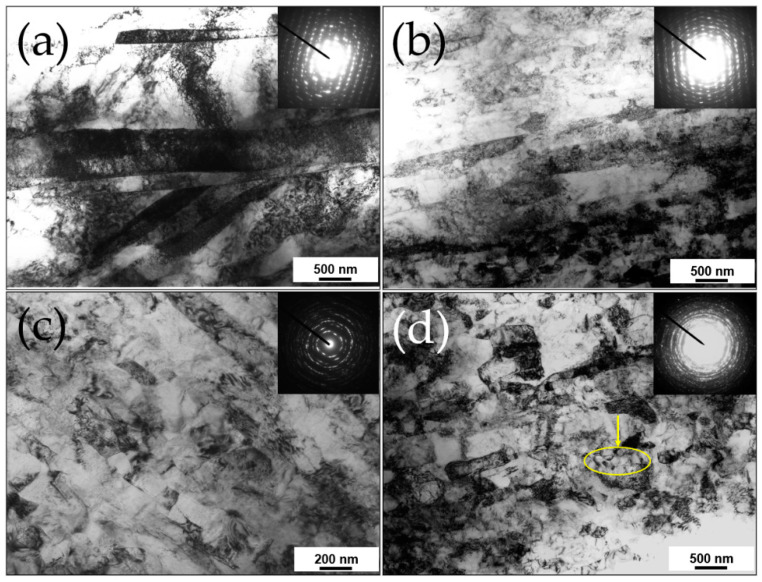
TEM micrographs and selected area diffraction (SAED) patterns of the rolled Ti at various SRs of (**a**) one, (**b**) three, (**c**) four and (**d**) five [41]. Reproduced with permission from Kim et al., *Scr. Mater.*; published by Elsevier, 2011.

**Figure 6 materials-13-04159-f006:**
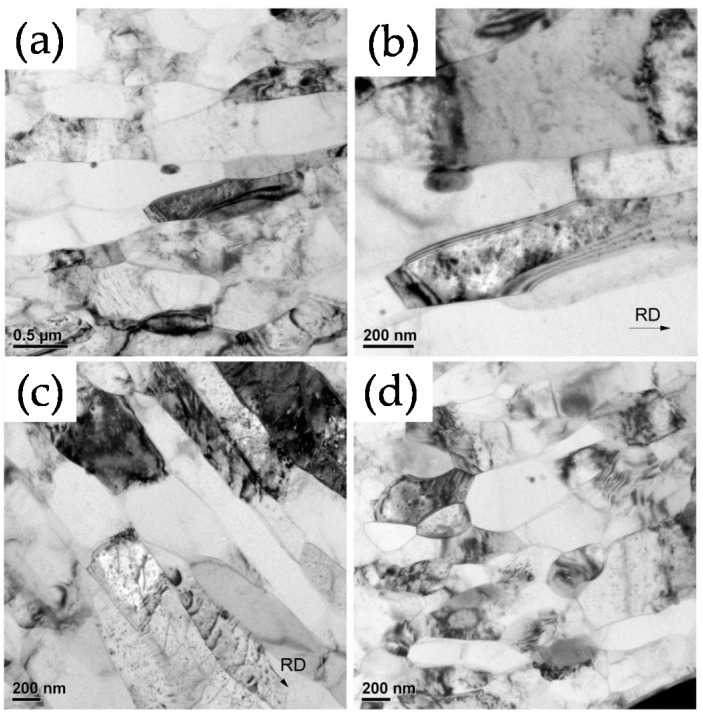
TEM micrographs of Al1050 alloy after asymmetric cryo-rolling: (**a**) and (**b**) SR of 1.1 for rolling direction (RD) at two different magnifications, (**c**) SR of 1.4 for RD, (**d**) SR of 1.4 for TD (RD: rolling direction; TD: transverse direction) [43].

**Figure 7 materials-13-04159-f007:**
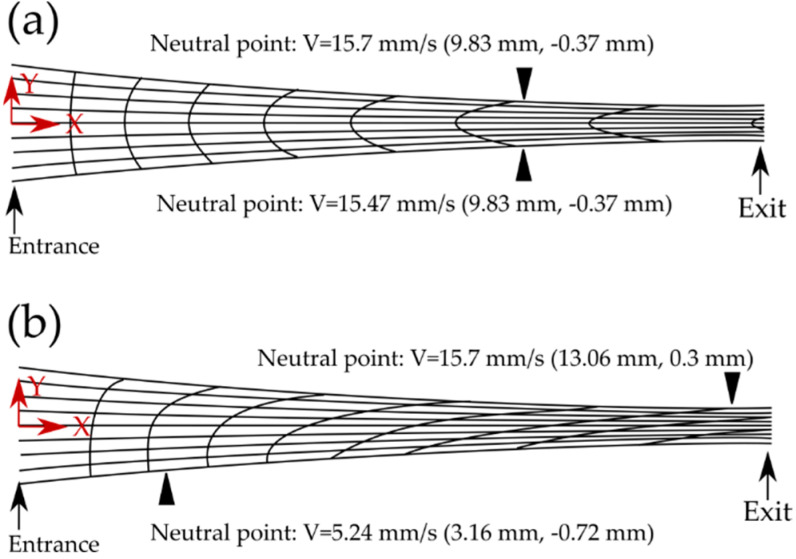
Comparison of grid distortions and neutral points: (**a**) ESR and (**b**) HRDSR [45]. Reproduced with permission from Ji et al., *Mater. Sci. Eng. A*; published by Elsevier, 2007.

**Figure 8 materials-13-04159-f008:**
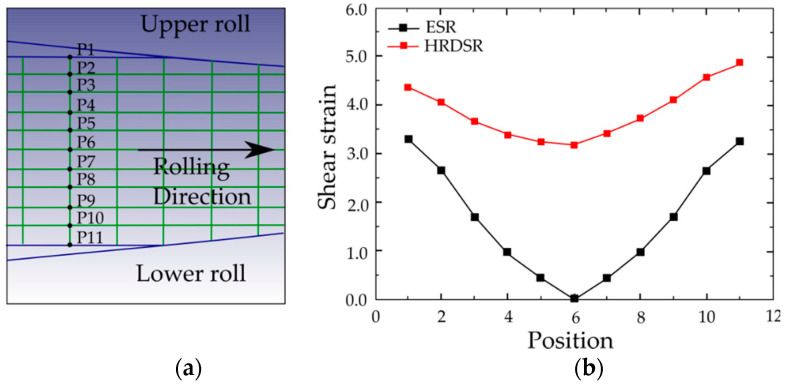
(**a**) The analyzed points in the enmeshed domain of simulation and (**b**) the accumulated shear strain along the thickness of the sheet using finite element simulation for HRDSR with SR = 3 for a thickness reduction of 70% by a single pass [46]. Reproduced with permission from Kim et al., *Mater. Sci. Eng. A*; published by Elsevier, 2013.

**Figure 9 materials-13-04159-f009:**
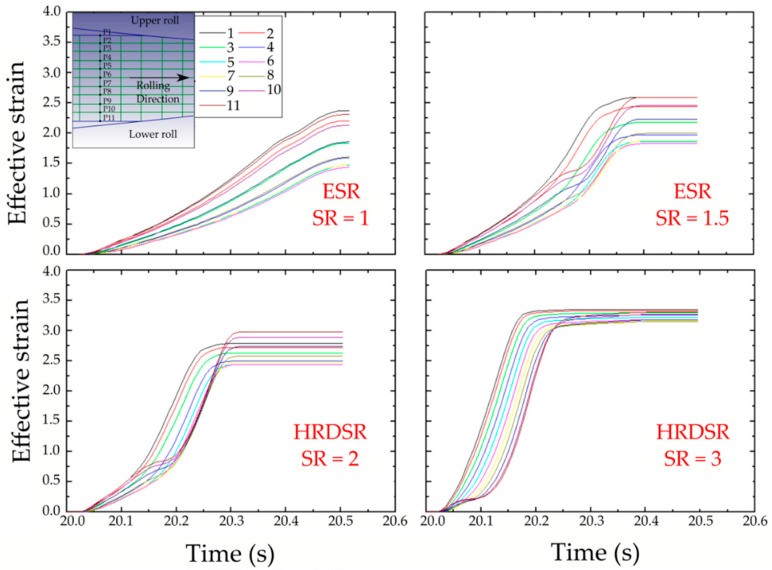
The accumulated effective strain along the thickness of the sheet during passage of the sheet material through the rolls, which is calculated as a function of processing time (in seconds) using finite element simulation for rolling with different SRs for a thickness reduction of 70% by a single pass under a high friction condition (the friction factor at the roll/sheet interface = 0.95) [47].

**Figure 10 materials-13-04159-f010:**
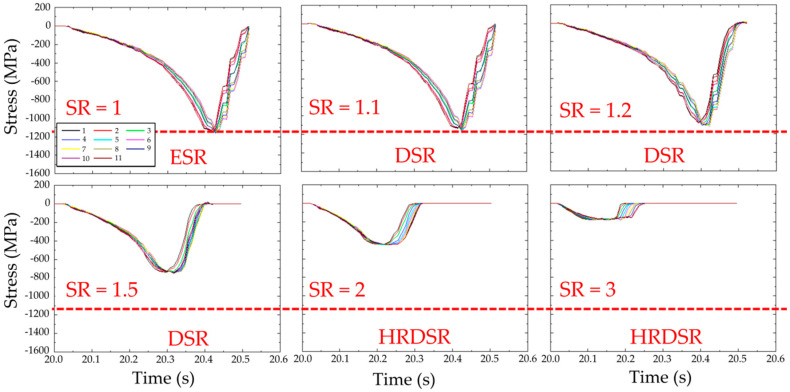
The effect of SR on the roll stress during passage of the sheet material through the rolls. The analyzed points in the enmeshed domain of simulation (Figure 8) and the roll stress along the thickness of the sheet using finite element simulation for rolling with different SRs for a thickness reduction of 70% by a single pass under a high friction condition (the friction factor at the roll/sheet interface = 0.95) [47].

**Figure 11 materials-13-04159-f011:**
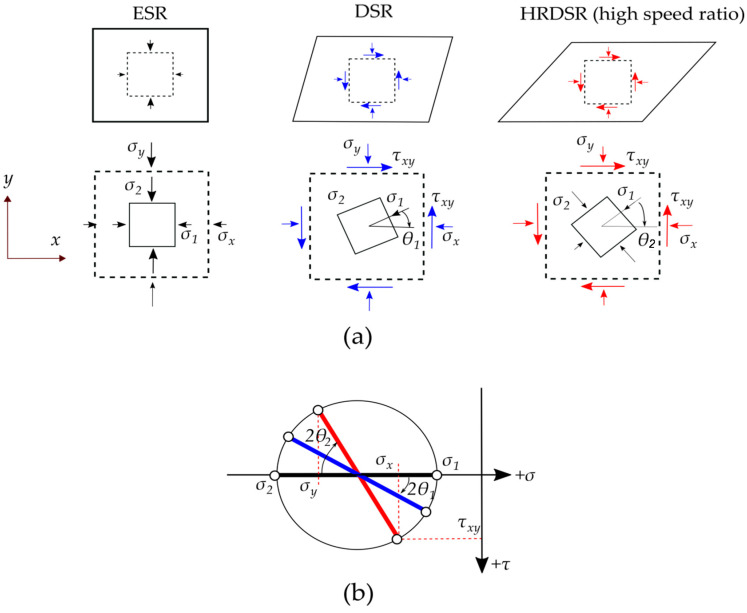
(**a**) Deformation of a square element after ESR, differential speed rolling (DSR), and HRDSR processes. (**b**) Mohr’s circle of stress representing the stress states corresponding to ESR, DSR, and HRDSR [49]. Reproduced with permission from Jeong et al., *Mater. Sci. Eng. A*; published by Elsevier, 2019.

**Figure 12 materials-13-04159-f012:**
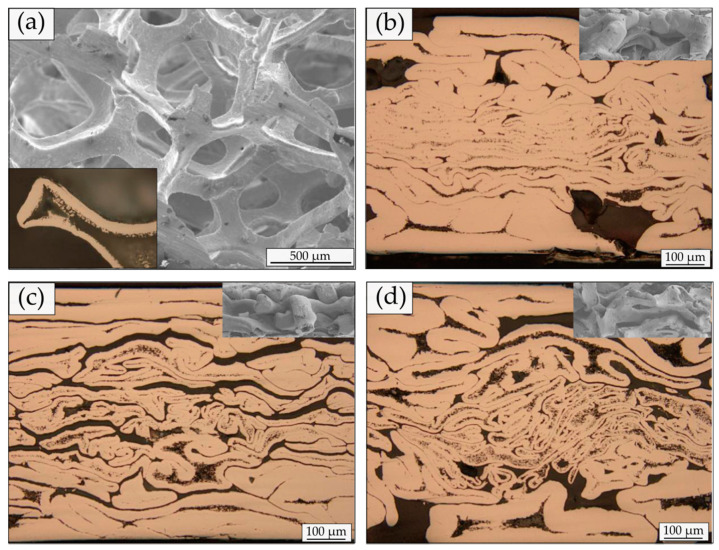
Optical photographs of the (**a**) as-received open-cell Ni foam, (**b**) ESR Ni, (**c**) HRDSR Ni (SR of two) and (**d**) HRDSR Ni (SR of three) samples. The inset in (**a**) shows the cross-section of the cell ligament. The insets in (**b**–**d**) show the SEM pictures taken on the lateral side of the rolled samples [49]. Reproduced with permission from Jeong et al., *Mater. Sci. Eng. A*; published by Elsevier, 2019.

**Figure 13 materials-13-04159-f013:**
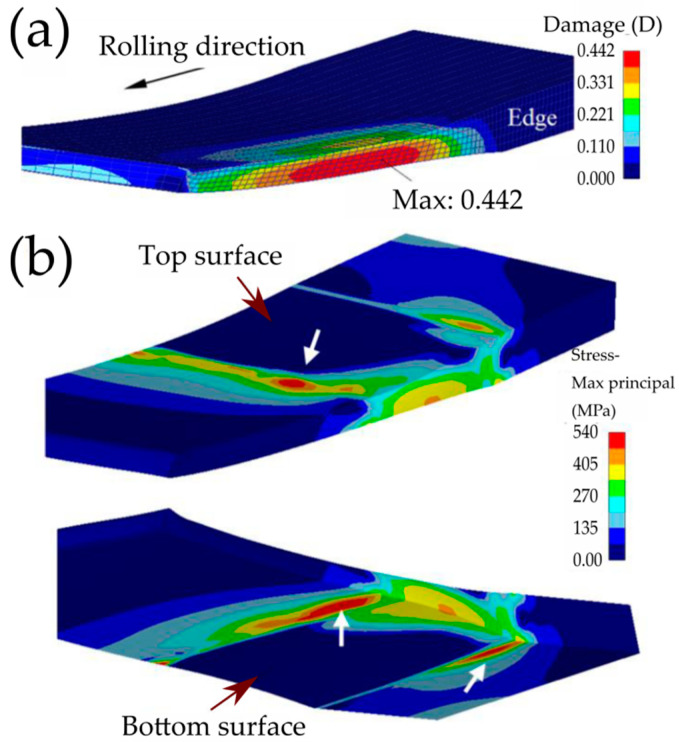
(**a**) Field of the material damage (*D*) and (**b**) tensile stresses developed during HRDSR (friction coefficient is 0.4; thickness reduction is 50%; rolling speed is 1.0 m/s; SR is 1.5) [38].

**Figure 14 materials-13-04159-f014:**
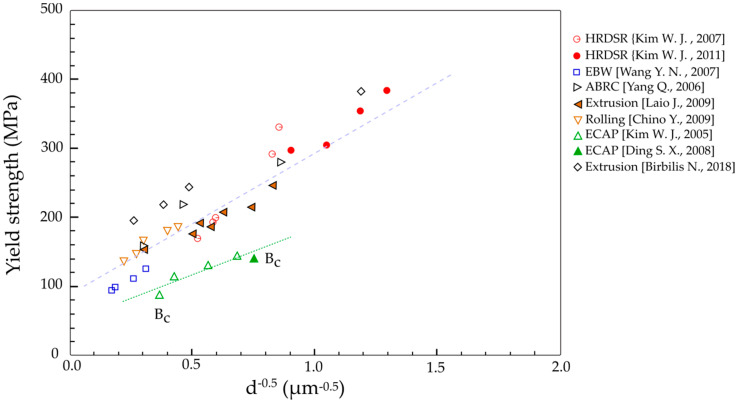
The 0.2% proof stress is plotted against grain size for the extruded, symmetrically rolled: ECAPed and HRDSR processed AZ31 alloys [33,35,63,64,65,66,67].

**Figure 15 materials-13-04159-f015:**
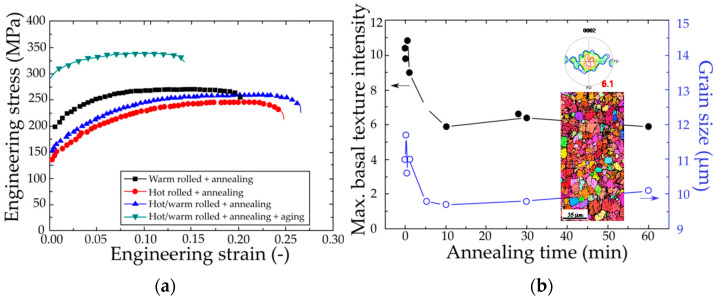
(**a**) Stress–strain curves of the rolled and heat-treated ZK60 alloys [61] and (**b**) the maximum intensity of basal texture and grain size of the hot/warm rolled alloy relative to the annealing time [61]. The inset in (**b**) shows the (0002) pole figure and EBSD inverse pole figure map of the hot/warm rolled plus annealed alloy. Reproduced with permission from Kim et al., *Mater. Sci. Eng. A*; published by Elsevier, 2014.

**Figure 16 materials-13-04159-f016:**
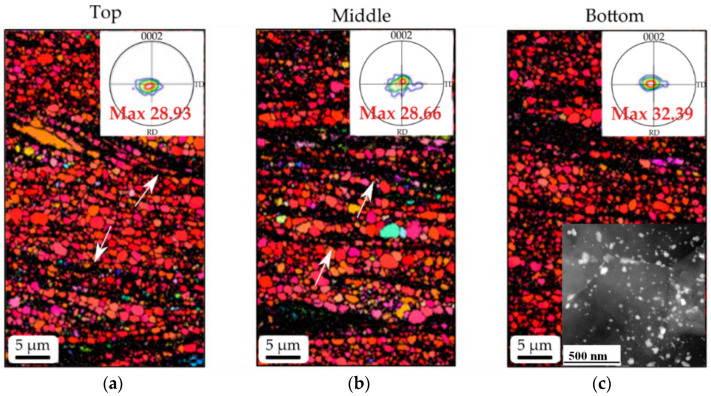
EBSD of the (**a**) top, (**b**) bottom and (**c**) middle of the HRDSRed ZK60 alloy, showing uniform microstructure and texture with narrow shear bands (white arrows) [83]. The inset in (**c**) shows the dispersion of fine Mg_2_Zn particles in the matrix. Reproduced with permission from Kim et al., *Scr. Mater.*; published by Elsevier, 2010.

**Figure 17 materials-13-04159-f017:**
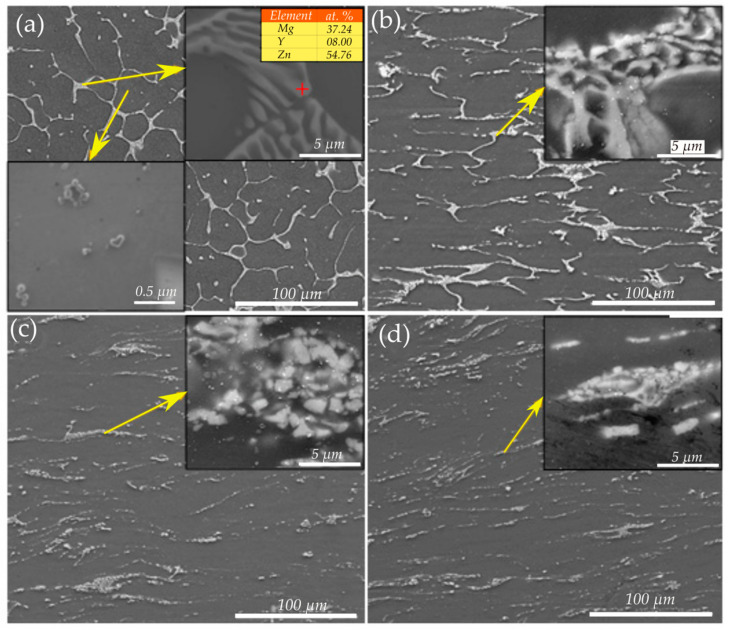
SEM micrographs of the (**a**) as-cast, (**b**) ESRed, (**c**) HRDSRed (with an SR of 2) and (**d**) HRDSRed (with an SR of 3) Mg–9.25Zn–1.66Y alloy. The insets in (**a**–**d**) show the eutectic *I*-phase pockets before and after ESR or HRDSR [84]. Reproduced with permission from Kwak et al., *Scr. Mater.*; published by Elsevier, 2015.

**Figure 18 materials-13-04159-f018:**
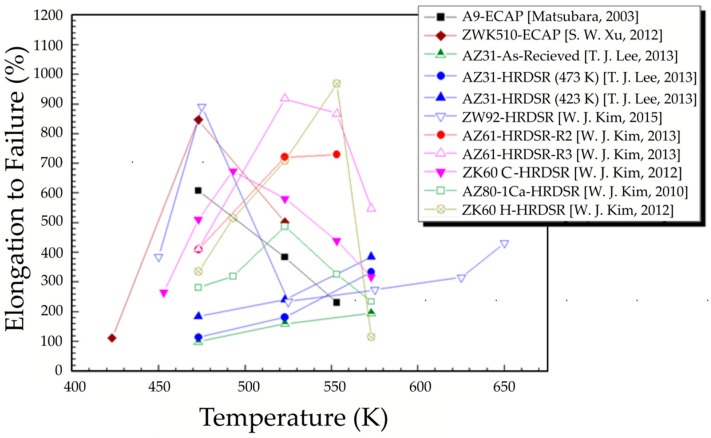
Elongation to failure vs. testing temperature of various Mg alloys at strain rate of 10^−3^ s^−1^ [85,86,87,88,89,90,91].

**Figure 19 materials-13-04159-f019:**
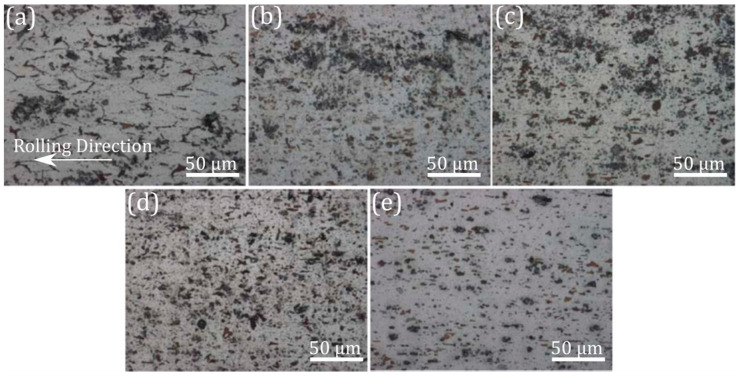
Optical micrographs of Al-TiC composite, (**a**) 10, (**b**) 12 and (**c**) 14 passes of conventional rolling and (**d**) first and (**e**) second pass of HRDSR [46]. Reproduced with permission from Kim et al., *Mater. Sci. Eng. A*; published by Elsevier, 2013.

**Figure 20 materials-13-04159-f020:**
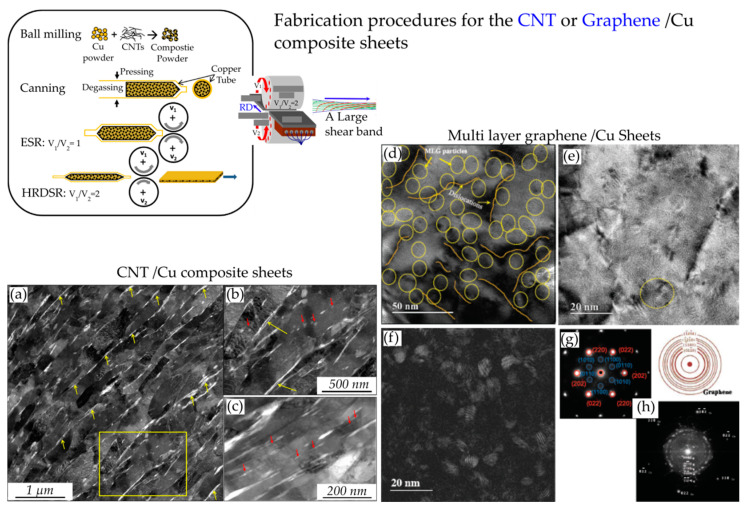
(**a**) A bright-field TEM micrograph of the 3% carbon nanotube- (CNT)-HRDSR Cu composite [74]. (**b**) High magnification TEM image of the box area in (**a**). (**c**) CNT in the grain interiors of the composite. (**d**,**e**) High-resolution TEM micrographs of the 1% multilayer graphene (MLG)-HRDSR Cu composite sowing the nanosized MLG particles in grain interiors [93]. (**f**) A dark field image of the microstructure shown in (**e**). (**g**) A selected diffraction pattern from the circle region in (**e**). (**h**) The diffraction pattern derived from the fast-Fourier transform (FFT) of the lattice image in (**e**). Reproduced with permission from Yoo et al., *Carbon*; published by Elsevier, 2013. Reproduced with permission from Kim et al., *Carbon*; published by Elsevier, 2014.

**Figure 21 materials-13-04159-f021:**
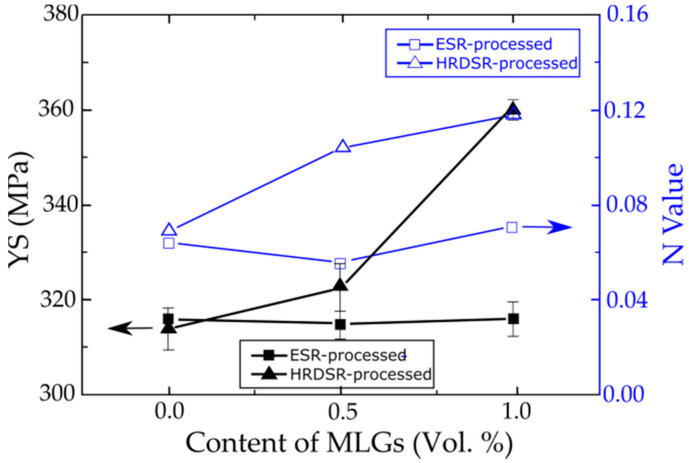
YS and *N* values versus MLG contents for the ESR- and HRDSR-processed Cu composites [93]. Reproduced with permission from Kim et al., *Carbon*; published by Elsevier, 2014.

**Figure 22 materials-13-04159-f022:**
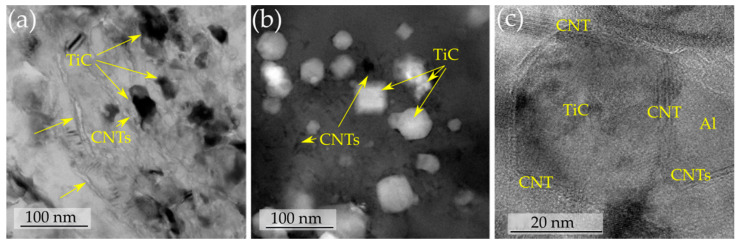
(**a**) Bright-field, (**b**) dark-field and (**c**) high-resolution TEM of Al/TiC/CNT composites [94]. Reproduced with permission from Kim et al., *Scr. Mater.*; published by Elsevier, 2014.

**Table 1 materials-13-04159-t001:** Literature review of the mechanical properties of various alloys subjected to HRDSR compared to the as-received samples. Here, GS is the grain size, El., YS and UTS are the tensile elongation, yield strength and ultimate strength, respectively, and Hv is the Vickers hardness.

As-Received							After HRDSR						
Alloy	Process	GS (µm)	El.%	YS (MPa)	UTS (MPa)	Hv	Processing Conditions	GS (µm)	El.%	YS (MPa)	UTS (MPa)	Hv	Ref.
AZ31	Extruded	30–60	-	164	-	-	T: 433 K Thick. red.: 70%	1.5	22	317	325	-	[35]
AZ31	Extruded	20	16	225	270	-	SR: 2 T: 423 K Water Quenched	0.6	7	382	405	-	[33]
Mg-Mn	Cast	95	-	-	-	53	SR: 3 T: 423 K	4	-	-	-	80	[68]
Pure nickel		35	35	~210	305	108	SR: 4	4	2	630	630	279	[69]
AZ91	Extruded	30–40	-	-	-	~80	SR: 3 T: 473 K	0.3–0.5	9	327	394	~95	[70]
Pure Cu		13.7	-	220	225	~122	Thick. red.: 65% T: Room	0.82	~40	~422	464	~153	[71]
6061 Al	Plate–Slow cooled in furnace to anneal	-	41	90	110	41	SR: 3 T: 413 K Thick. red.: 70% (Aged)	0.37	~6	455	485	145	[72]
Pure Al		35	~17	~110	100	-	DSR (90%)	0.5	~4	250	250		[73]
							ESR (90%)		~6	155	155	-	
5052 Al	50 Sheets	~95	65	137	32	-	4 Pass; SR: 4	0.7	4.2	380	390	-	[28]
Composites Cu + 3%CNT	-	-	-	-	-	-	HRDSR: SR: 2; T: 473 K	0.6	417	500	12.2	135	[74]
							ESR	1.5	363	413	17.8	115	

**Table 2 materials-13-04159-t002:** Comparison between the mechanical properties of pure Ti obtained from HRDSR and other processing routes. Here, GS is the grain size, El., YS and UTS are the tensile elongation, yield strength and ultimate strength, respectively, and Hv is the Vickers hardness.

Processing Conditions	GS (µm)	El. %	YS (MPa)	UTS (MPa)	Hv	Ref.
As-received	10	~35	403	532		[76]
HRDSR- Thick. red.: 63%.-1P	0.3		780	895	
HRDSR-Thick. red.: 63%.-1P-Anneal(1h)		19%	804	915	
Hot Roll (Rod)	10	24	440	480	183.5	[77]
ECAP (773–723 K) 7P	0.3	16	520	540	239.6
ECAP (773–723 K) 7P/HPT (723 K)		30	530	640	275.3
ECAP (773–723 K) 7P/HPT (293 K)		25	625	730	318.1
As-received (Hot rolled)(Ti-grade 2)	30–60	30	354	487		[75,78]
ECAP-8P		27	549	633	
ECAP-10P	0.45	20.6	582	645	
ECAP-8P+Cold roll (77%) (grade 2)		17.6	665	938	
ECAP-10P+Cold roll (77%) (grade 2)	0.19	14.5	736	945	
ECAP-10P + Hot roll (81%) (grade 2)		18	736	928	
Cold roll (77%) (Ti-grade 4)		15.2	650	791	
As-received		25	630	740	
ECAP-8P		19	750	815	
ECAP (8) + Cold roll (83%)		10.7	1006	1135	
CR (80%)		10		1130	
As-received (Ti-VT1-0)	10	27	380	460	
ECAP (8)	0.28	14	640	710	
ECAP (8) + Cold roll (73%)		12.5	940	1037	
ARB	0.09	37	720	900		[79]
ARB-Ann. 373 K	0.1	22	710	880	
ARB-Ann. 473 K	0.12	17	702	870	
ARB-Ann. 573 K	0.14	12	700	840	
ARB-Ann. 673 K	0.27& 0.50	11	580	780	
ARB-Ann. 723 K	0.33& 0.70	13	550	680	
ARB-Ann. 773 K	2.3	8	500	520

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
