# Peer review of "Effect of Grain Refinement and Dispersion of Particles and Reinforcements on Mechanical Properties of Metals and Metal Matrix Composites through High-Ratio Differential Speed Rolling"

_materials, 2020, doi:10.3390/ma13184159_

Round 1
Reviewer 1 Report
The research results presented in the article are interesting. Step by step authors presents results in reasonable way. The following considerations are required to be considered:
68: To show the difference in the quality of the sheets obtained with the HRDSR and ESR methods, the authors should present the results of e.g. surface roughness or thickness deviation of the sheets
100 – 104: The authors should provide a method of grain size measurement and show the place of sampling from metal sheets for metallographic processes
125: The authors should mark the described areas on the figure
157 end 166: There are no references to literature. Is this own research?
165: In Figure 9 the graph is described by "Stress", in the description is "force". Units are also missing
265: The presentation of the deformation of the AZ31 sheet on the basis of the tensile test is not reliable. The correct way is to present Forming Limit Curves (FLC), which represent the formability of sheet metal in stamping processes. Therefore, the authors should additionally present FLC diagrams for AZ31 alloy sheet obtained in the discussed processes.
Thre refereces below the paper are missing, please add it
Author Response
Comment 1:
68: To show the difference in the quality of the sheets obtained with the HRDSR and ESR methods, the authors should present the results of e.g. surface roughness or thickness deviation of the sheets.
We do not have the results about surface roughness or thickness deviation of the sheets. The text is revised and some discussion on the damage of surface are added in the revised paper: “ Comparing the out-coming sheet from HRDSR and equal speed rolling (ESR), the sheet after HRDSR has a higher quality in flatness and shape. The HRDSR-processed sheet is flat and smooth while the ESR-processed sheet is wavy and uneven when a large thickness reduction per pass is applied (Figure 2d). Pesin et al. [38, 39] showed in the finite element method simulation that high shear strain through thickness of the sheet can only be achieved in HRDSR when the friction coefficient between the sample and the surface of the rolls is high, but the high friction can cause material damage and roll wear. Through control of the amount of thickness reduction per pass and addition of a small amount of lubricants, however, there have been efforts to minimize the material damage and wear problems in the continuous HRDSR mill (Figure 2(c)).
Comment 2:
100 – 104: The authors should provide a method of grain size measurement and show the place of sampling from metal sheets for metallographic processes.
The average grain size has been measured using electron back-scattered diffraction (EBSD) analysis with the aid of TSL-OIM software. This is mentioned in the revised paper. The place of sampling from metal sheets for metallographic processes is mentioned in the caption of Figure 3 : “The EBSD IQ maps at the middle of longitudinal cross sections of the sheet”
Comment 3:
125: The authors should mark the described areas on the figure
Figure 5d was revised with adding an arrow as mark.
Comment 4:
157 end 166: There are no references to literature. Is this own research?
This part was from the un-published work by W.J. Kim. This was referred to as that in the revised paper.
Comment 5:
165: In Figure 9 the graph is described by "Stress", in the description is "force". Units are also missing
Unit (MPa) was given in the revised paper. Force was replaced by stress in the text.
Comment 6:
265: The presentation of the deformation of the AZ31 sheet on the basis of the tensile test is not reliable. The correct way is to present Forming Limit Curves (FLC), which represent the formability of sheet metal in stamping processes. Therefore, the authors should additionally present FLC diagrams for AZ31 alloy sheet obtained in the discussed processes.
We could not find the FLC data of the HRDSRed Mg alloys anywhere including ours. Instead, the following words and references are added in the revised paper: “The results showed the possibility of producing Mg alloy sheets with high formability at room temperature with thanks of texture weakening, and high strength through the aging process after sheet forming. Huang et al. [62] showed that the DSRed AZ31 Mg alloy sheet with weakened basal texture exhibited a larger tensile elongation, a smaller Lankford value, a larger strain hardening exponent and improved Erichsen values at room temperature. The HRDSR-processed AZ31 Mg alloy also exhibited a significant improvement of the limit dome height by 46% compared to the commercial AZ31 sheet at 523 K [37].”
Comment 7:
Thre refereces below the paper are missing, please add it.
The proper references were added in the revised paper.
Reviewer 2 Report
The review titled “Effect of grain refinement and dispersion of particles on mechanical properties of metals through high-ratio differential speed rolling” reported an overview of different works of a severe plastic deformation process named "high-ratio differential speed rolling (HRDSR)" with the aim to explain its effects on the metals mechanical properties. In my opinion, as motivated point by point in the following, an extensive revision of the text is needed. English editing is required in some parts of the text.
I observe that 39% of the references are related to the author Woo Jim Kim. Furthermore, more than 90% of the text is based on the works of this author. This fact gives the impression that there are no more researchers in the world working on this field, and that this review is not objective. Moreover, among the 89 references only 17 have been published in the last four years, and three of them are directly related to the subject of the manuscript. I propose to change the text of the manuscript, including the most recent results of other authors in the DSR and HRDSR field, to explain each of the sections of this manuscript. You can see and use, for example, the publications of J.J. Park, A. Pesin, D. Pustovoytov and others.
In order to maintain an equality of the units throughout the text, I propose to change the temperature unit of figure two to Kelvin.
At the beginning of each paragraph, where the results of an investigation are cited, it is necessary to remain the main author' surname of cited work. For example, in line 82, first paragraph of section 2.1, it is not clear who carried out the cited experiment. In line 129, first paragraph of section 2.2., and other paragraphs along the text, it is possible to see the same mistake. Throughout the review, the authors' surnames are reminded few times and principally in relation to W.J.Kim.
I want to see a explanation of the picture 5d, in which a combination of SR4 and SR5 was used.
In line 119, the authors wrote... "Nanosized subgrains or cells with a size of ∼0.1 μm also developed within some of the grains (indicated by arrows)", but I could'n find the arrows. Please, edit the text or the picture.
Figure 6a and 6 b are related to the ESR and HRDSR, respectively. Please, edit the order of these processes in line 133. Moreover, the paragraph in lines 132-135 must be edited for a better understanding.
I proposed to write a detailed explanation of figures 8 and 9,
and indicate where these drawings were obtained from. Furthermore, it is not clear for me, why the unit in the axis "x" of figure 8 is seconds, when in the text the authors wrote about "the strain distribution along the thickness". I would recommend writing in each picture of figure 9, the name of the used processes (ESR, DSR, and HRDSR).
In figure 10, for the HRDSR procaces is necessary to edit the angle to "theta 2".
I suggest to write the paragraph 279-288 among the tables 1 and 2.
In the end of tables 1 and 2 must be the explanation of abbreviations: GS, EI,YS, UTS.
Along the table 1 maintain the form "Thick Red.: XX%".
What is the mean of the temperature T: 423 K WQ? Please, correct or explain it.
Please, check throughout the manuscript that the authors' surnames are well written and capitalized: for instance Stolyarov (line 284) and reference 21, Bahmani (line 470).
Author Response
Comment 1:
The review titled “Effect of grain refinement and dispersion of particles on mechanical properties of metals through high-ratio differential speed rolling” reported an overview of different works of a severe plastic deformation process named "high-ratio differential speed rolling (HRDSR)" with the aim to explain its effects on the metals mechanical properties. In my opinion, as motivated point by point in the following, an extensive revision of the text is needed. English editing is required in some parts of the text.
English was polished and more carefully edited.
Comment 2:
I observe that 39% of the references are related to the author Woo Jim Kim. Furthermore, more than 90% of the text is based on the works of this author. This fact gives the impression that there are no more researchers in the world working on this field, and that this review is not objective. Moreover, among the 89 references only 17 have been published in the last four years, and three of them are directly related to the subject of the manuscript. I propose to change the text of the manuscript, including the most recent results of other authors in the DSR and HRDSR field, to explain each of the sections of this manuscript. You can see and use, for example, the publications of J.J. Park, A. Pesin, D. Pustovoytov and others.
I am sorry for extensive referring to my own works. This was inevitable because the topic for this paper for severe plastic deformation by HR(DSR). The papers related to ordinary DSR (the related review paper is elsewhere) that do not produce SPD ultrafine grains were excluded in this work. In the revised paper, the HRDSR works conducted by A. Pesin, D. Pustovoytov and W. Polkowski et al. were referred and the figures (Figure 6 and Figure 13) related to their works were added.
Comment 3:
In order to maintain an equality of the units throughout the text, I propose to change the temperature unit of figure two to Kelvin.
The unit was changed as suggested.
Comment 4:
At the beginning of each paragraph, where the results of an investigation are cited, it is necessary to remain the main author' surname of cited work. For example, in line 82, first paragraph of section 2.1, it is not clear who carried out the cited experiment. In line 129, first paragraph of section 2.2., and other paragraphs along the text, it is possible to see the same mistake. Throughout the review, the authors' surnames are reminded few times and principally in relation to W.J.Kim.
The name of author/researcher was added in some missing points.
Comment 5:
I want to see a explanation of the picture 5d, in which a combination of SR4 and SR5 was used.
In line 119, the authors wrote... "Nanosized subgrains or cells with a size of ∼0.1 μm also developed within some of the grains (indicated by arrows)", but I could'n find the arrows. Please, edit the text or the picture.
The arrow mark was added to the Figure 5.
Figure 5d was removed to avoid confusion. Figure 5e now became Figure 5d.
Comment 6:
Figure 6a and 6 b are related to the ESR and HRDSR, respectively. Please, edit the order of these processes in line 133. Moreover, the paragraph in lines 132-135 must be edited for a better understanding.
The order was corrected and following text was added to emphasize the order of ESR and HRDSR in the Figure. “Figure 7 shows the grid distortions and neutral points for ESR and HRDSR processes.”
Comment 7:
I proposed to write a detailed explanation of figures 8 and 9,
and indicate where these drawings were obtained from. Furthermore, it is not clear for me, why the unit in the axis "x" of figure 8 is seconds, when in the text the authors wrote about "the strain distribution along the thickness". I would recommend writing in each picture of figure 9, the name of the used processes (ESR, DSR, and HRDSR).
The following words were added in the text or figure captions.
>This is shown in Figure 9 where the effective strain accumulated during rolling is monitored as a function of processing time [47].
>Furthermore, the accumulated strain is more uniformly distributed along the thickness of the sample after HRDSR.
>Figure 10 shows the effect of SR on the roll stress, which is monitored as a function of processing time, under a thickness reduction of 70% by a single pass [47]. The maximum roll stress tends to decrease with increase of SR.
>Figure 9. The accumulated effective strain along the thickness of the sheet during passage of the sheet material through the rolls, which is calculated as a function of processing time (in seconds) using finite element simulation for rolling with different SRs for a thickness reduction of 70% by a single pass under a high friction condition (the friction factor at the roll/sheet interface = 0.95) [47].
>Figure 10. The effect of SR on the roll stress during passage of the sheet material through the rolls. The analyzed points in the enmeshed domain of simulation (Figure 8) and the roll stress along the thickness of the sheet using finite element simulation for rolling with different SRs for a thickness reduction of 70% by a single pass under a high friction condition (the friction factor at the roll/sheet interface = 0.95) [47].
The names of ESR, DSR, and HRDSR were indicated in Figure 9 and 10 as advised.
Comment 8:
In figure 10, for the HRDSR procaces is necessary to edit the angle to "theta 2".
This was fixed.
Comment 9:
I suggest to write the paragraph 279-288 among the tables 1 and 2.
This can be done in final editing process.
Comment 10:
In the end of tables 1 and 2 must be the explanation of abbreviations: GS, EI,YS, UTS.
The following text was added:
“Here, GS is the grain size, El., YS and UTS are the tensile elongation, yield strength and ultimate strength, respecively, and Hv is the Vickers hardness.
”
Comment 11:
Along the table 1 maintain the form "Thick Red.: XX%".
All of the thickness reduction phrases in the tables were changed to "Thick Red.: XX%".
Comment 12:
What is the mean of the temperature T: 423 K WQ? Please, correct or explain it.
WQ was changed to water quenched.
Comment 13:
Please, check throughout the manuscript that the authors' surnames are well written and capitalized: for instance Stolyarov (line 284) and reference 21, Bahmani (line 470).
It was checked and the mentioned authors’ surnames were capitalized.
Reviewer 3 Report
Summary
This paper reviews the literature on Differential speed rolling and in particular high-ratio DSR (HRDSR) techniques. It also compares with other common severe plastic deformation techniques in terms of impact on microstructure and mechanical properties.
Comments
The paper is comprehensive and written in good English. This is presented as a review paper and it happens that over a third of all references are previous work from the authors. It seems like a lot of self-citation although this is a review paper. Please consider carefully if all the litterature on the subject is well represented. And please also keep in mind this amount of self-citation is not always well received.
As the authors are experts on the topic, I have not many comments on the scientific soundness of the work which is well structured and easy to read
In the first part of the section 2.1 (line 80-97), it is unclear whether this information comes out of the authors’ own work or is taken from literature as no references are given in the text. It is the same in section 2.2, is the finite element simulation the result of the authors’ own work? Please make sure that everything coming from previous work is referenced accordingly in the text and not only on figures.
Line 32, I am not a bit fan of the semantic “Manufacturing types”. Maybe you could use “processes” instead of “types”?
Author Response
Comment 1:
The paper is comprehensive and written in good English. This is presented as a review paper and it happens that over a third of all references are previous work from the authors. It seems like a lot of self-citation although this is a review paper. Please consider carefully if all the literature on the subject is well represented. And please also keep in mind this amount of self-citation is not always well received.
I am sorry for extensive referring to my own works. This was inevitable because the topic for this paper for severe plastic deformation by HR(DSR). The papers related to ordinary DSR (the related review paper is elsewhere) that do not produce SPD ultrafine grains were excluded in this work. In the revised paper, the HRDSR works conducted by A. Pesin, D. Pustovoytov and W. Polkowski et al. were referred and the figures (Figure 6 and Figure 13) related to their works were added.
Comment 2:
As the authors are experts on the topic, I have not many comments on the scientific soundness of the work which is well structured and easy to read
Comment 3:
In the first part of the section 2.1 (line 80-97), it is unclear whether this information comes out of the authors’ own work or is taken from literature as no references are given in the text. It is the same in section 2.2, is the finite element simulation the result of the authors’ own work? Please make sure that everything coming from previous work is referenced accordingly in the text and not only on figures.
As you mentioned, the citation was unfortunately missed in the text. The related references were added.
Comment 4:
Line 32, I am not a bit fan of the semantic “Manufacturing types”. Maybe you could use “processes” instead of “types”?
This was revised as advised.
Reviewer 4 Report
In this paper, the authors aimed to provide a critical review of the differential speed rolling and its roles in altering the mechanical properties and microstructures of various metals. The information will be interesting in the field in terms of the benefits of this technique. A couple of comments are as follows:
- In section 2.1: What is the rolling speed of the roller for various ratios? Is it fixed or not? This information is absent in this part.
- As a review paper, it is better to include the authors’ own viewpoint on the future direction of the technique or any critical issues that need to tackle in the future. A short paragraph in terms of perspectives before the conclusions is suggested.
Author Response
Comment 1:
In section 2.1: What is the rolling speed of the roller for various ratios? Is it fixed or not? This information is absent in this part.
This information was given in the caption of Figure 3: Figure 3. The EBSD IQ maps at the middle of longitudinal cross sections of the sheet (rolling direction (RD) is the normal direction of figures plane) with SR of (a) 1.0, (b) 1.1, (c) 1.2, (d) 1.5, (e) 2.0 and (f) 3.0 [40]. The speed ratio between the upper and lower rolls varied from 1 to 3 with the speed of lower roll fixed as 3 rpm (Roll diameter of 400 mm).
Comment 2:
As a review paper, it is better to include the authors’ own viewpoint on the future direction of the technique or any critical issues that need to tackle in the future. A short paragraph in terms of perspectives before the conclusions is suggested.
In the conclusion part, the following words are added in the revised paper: “4. HRDSR has a high commercial application potential due to the possibility of scaling the product up to commercial dimensions and continuous production of the product, but there are many parameters (such as contact friction, thickness reduction per pass, the roll speed ratio, the roll speed and temperature, and the sheet temperature) to be optimized for achieving the very fine-grained sheets with controlled texture, small surface roughness and uniform thickness distribution.
”
Round 2
Reviewer 1 Report
Thanks for the answers. I have no more comments
Author Response
Thanks for the answers. I have no more comments
>>Thank you.
Reviewer 2 Report
The review titled “Effect of grain refinement and dispersion of particles on mechanical properties of metals through high-ratio differential speed rolling” reported an overview of different works of a severe plastic deformation process named "high-ratio differential speed rolling (HRDSR)" with the aim to explain its effects on the metals mechanical properties.
After the first revision of the text, the authors have corrected most of the comments. Despite this, it is necessary to make minor corrections before the publication.
1 - In line 119, the authors wrote... "Figure 6 a–e show the TEM micrographs and the corresponding selected area diffraction (SAED) patterns of the pure Ti deformed by HRDSR ……”. Nevertheless, the figure 6 is related to the alloy Al1050. Moreover, the figure 6e was removed, but the text was not edited. Please, edit the text or the picture.
2 – References must be written in accordance with the journal's instructions that have been published especially for authors at the following link
https://www.mdpi.com/journal/materials/instructions#preparation
Please, go to the indicated link and check the point "Back Matter" where a clear explanation of how to write the references is given.
3 – The point of excessive self-citation remains open.
Author Response
1 - In line 119, the authors wrote... "Figure 6 a–e show the TEM micrographs and the corresponding selected area diffraction (SAED) patterns of the pure Ti deformed by HRDSR ……”. Nevertheless, the figure 6 is related to the alloy Al1050. Moreover, the figure 6e was removed, but the text was not edited. Please, edit the text or the picture.
--> This error was fixed.
2 – References must be written in accordance with the journal's instructions that have been published especially for authors at the following link
https://www.mdpi.com/journal/materials/instructions#preparation
Please, go to the indicated link and check the point "Back Matter" where a clear explanation of how to write the references is given.
--> The format of references was corrected.
3 – The point of excessive self-citation remains open.
--> Thank you.